# NEURAL LIGHTING PRIORS FOR INDOOR SCENES

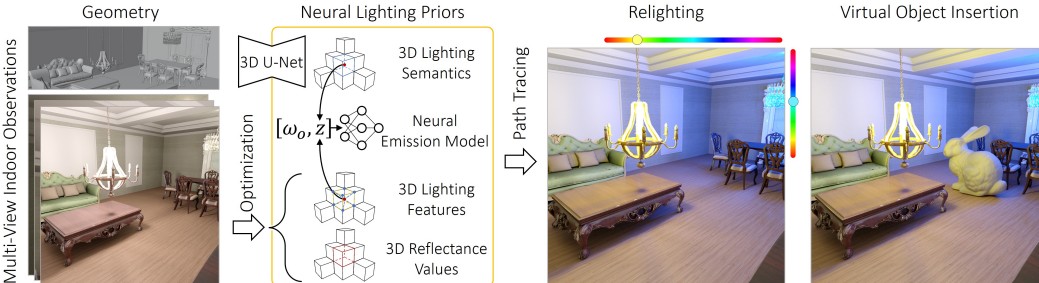

Figure 1: **Neural Lighting Priors.** We present Neural Lighting Priors for reconstructing a 3D neural surface emission field from sparse multi-view images. We represent the lighting of the scene with a neural emission model, locally conditioned on 3D lighting and semantic features. We use a coarse spatially varying representation and fit the local latent codes by re-rendering the scene using path tracing and optimizing the reconstruction loss. Our representation enables photo-realistic relighting and virtual object insertion even in a sparse setting.

## ABSTRACT

We introduce Neural Lighting Priors, a learned surface emission model for indoor scenes. Given multi-view observations as well as the geometry of a scene, we decouple spatially varying lighting and material parameters. Existing inverse rendering methods typically use hand-crafted emission models or require a large number of views to better constrain the highly ambiguous appearance decomposition task. We aim to overcome these limitations by introducing an expressive learned parametric emission model and utilizing semantic information to sufficiently constrain the optimization, thus allowing us to infer light sources, even if they are not visible in the observations. We model the emitted radiance with a neural field parameterized by the emitting direction and a local latent code stored in a voxel grid. At test time, we fit the local latent codes to the scene using differentiable path tracing, optimizing the reconstruction loss. Our reconstruction allows us to insert virtual objects in a scene and gives us control over the emitters to change their emission color and intensity. Thanks to the learned 3D prior, our method requires fewer views than state-of-the-art relighting methods, gives more control, and also improves the relighting quality.

## 1 INTRODUCTION

Precise estimation of lighting conditions holds paramount significance in a multitude of subsequent applications, notably within the realms of virtual and augmented reality (AR). Image observations contain the interaction of lighting and material. Our goal is to decouple the lighting from a sparse set of images given pre-scanned geometry. One possible application is an AR meeting room, where virtual participants need to be inserted into the scene photo realistically. While the room's geometry can be scanned and reconstructed once in advance, the lighting may change across sessions, motivating sparse view lighting estimation with known geometry. Previous methods directly optimize for emission parameters using inverse rendering (Maier et al., 2017; Azinovic et al., 2019; Nimier-David et al., 2021; Li et al., 2022; Barron & Malik, 2015), which gives explicit control over the scene lighting but mostly rely on hand-crafted priors for the lighting to constrain this highly underdetermined optimization problem. Recent methods achieve impressive relighting results using a large number of views (Philip et al., 2021; Wu et al., 2023; Yu et al., 2023).

Recently, learning-based methods have been applied to directly estimate complex lighting conditions (Gardner et al., 2017; 2019; Li et al., 2020; 2022; Zhu et al., 2022; Weber et al., 2022). Such methods are able to reconstruct high-quality incident illumination models to allow convincing virtual object insertion. However, they cannot provide consistent control over the scene's lighting and often require training on synthetic imagery, causing a domain gap (Wang et al., 2021; Li et al., 2020; Philip et al., 2021; Li et al., 2022; Zhu et al., 2022; Weber et al., 2022).

In this work, we combine an explicit inverse rendering method with the expressiveness of neural networks. We learn a neural parametric emission model from synthetic data, which can be fit to real scenes to facilitate both relighting and virtual object insertion. Our model utilizes semantic information to further constrain the optimization, which allows us to reconstruct high-quality emissions, and also infer light sources that are completely unobserved across all of the input images.

Specifically, we model the surface emission with a locally conditioned neural field (Xie et al., 2022). We render views from a large set of synthetic scenes with photo-realistic lighting conditions and train a generic emission model to represent various realistic emitters. At test time, we use a differentiable path tracer to reconstruct the observations and optimize the local lighting features. To enable reconstruction from sparse or even incomplete observations, we further condition on local features predicted from the scene's geometry via a 3D convolutional neural network indicating the likelihood that a certain piece of geometry is an emitter.

Our approach benefits from the advantages of physically-based inverse rendering and neural representations. First, explicit surface emission reconstruction enables lighting editing and virtual object insertion with consistent global illumination. Second, our learned emission model is capable of representing complex emission profiles with fine details and it is not limited by hand-crafted definitions. By leveraging a learned semantic prior, we additionally constrain the optimization to significantly reduce the required number of views and even infer light sources that are not directly observed in any of the input images. In summary, our main contributions are:

- We propose a neural-field-based lighting representation to model emitted radiance of surface points in conjunction with a voxel-based emitter sampling technique to efficiently render our neural representation.

- We introduce a learned prior for complex indoor lighting conditions leveraging semantical information to sufficiently constrain the highly ill-posed appearance decomposition task.

- We introduce high-quality textured mesh light sources to the 3D-Front dataset (Fu et al., 2021) and render 976 train 100 test scenes.

## 2 RELATED WORK

**Lighting Reconstruction.** Earlier light estimation methods for room-scale scenes focused on predicting the incident illumination from single images. Gardner et al. (2017); Wang et al. (2022a); LeGendre et al. (2019); Weber et al. (2022) predict global spherical environment maps. Gardner et al. (2019) proposed to approximate the environment map with Spherical Gaussians (SG) to reduce the task's complexity and achieve better generalization. Since these approaches use a global lighting representation, they are not able to reconstruct spatially varying (SV) lighting, which is crucial for room-scale scenes.

Srinivasan et al. (2020); Wang et al. (2021); Maier et al. (2017); Philip et al. (2021); Li et al. (2020) use local incident lighting representation to predict pixel, patch-wise or global environment maps approximated by Spherical Harmonics (SVSH) (Maier et al., 2017), SVSG (Li et al., 2020), irradiance maps (Philip et al., 2021), incident light fields (Yao et al., 2022; Zhang et al., 2023; Wang et al., 2023), or volumetric lighting (Choi et al., 2023; Wang et al., 2022b). They excel in reconstructing the lighting of a scene, but they cannot model consistent light transport prohibiting light editing.

Recent methods aim at decomposing the scene in a physically-based way using inverse path tracing (Azinovic et al., 2019; Nimier-David et al., 2021; Li et al., 2022; Whelan et al., 2016; Wu et al., 2023; Lin et al., 2024; Yu et al., 2023). They model the emitted radiance and provide globally consistent lighting with scene editing capabilities. Nevertheless, they rely on hand-crafted priors and emission models, such as mesh lighting with cosine emission profile (Azinovic et al., 2019; Nimier-David et al., 2021; Wu et al., 2023) or Spherical Gaussians (SG) (Li et al., 2022) to constrain the

| Method | Input | Object insertion | Lighting Reconstruction | | | | Relighting | | Physically-based Rendering | Lighting Representation |
|---|---|---|---|---|---|---|---|---|---|---|
| | | | Spatially-Varying | Surface Emission | Complex Emission Distribution | Geometry Prior | Light Insertion | Light Editing | | |
| **IndoorIllum** (Gardner et al., 2017) | Single | ✓ | ✗ | ✗ | ✗ | ✗ | ✗ | ✗ | ✗ | EnvMap |
| **DeepPara** (Gardner et al., 2019) | Single | ✓ | ✗ | ✗ | ✗ | ✗ | ✗ | ✗ | ✗ | EnvMap |
| **StyleLight** (Wang et al., 2022a) | Single | ✓ | ✗ | ✗ | ✗ | ✗ | ✗ | ✗ | ✗ | EnvMap |
| **Lighthouse** (Srinivasan et al., 2020) | Stereo | ✓ | ✓ | ✗ | ✗ | ✗ | ✗ | ✗ | ✗ | Lighting volumes |
| **Indoor3DSVL** (Wang et al., 2021) | Single | ✓ | ✓ | ✗ | ✗ | ✗ | ✗ | ✗ | ✗ | Lighting volumes |
| **Intrinsic3D** (Maier et al., 2017) | Multi | ✓ | ✓ | ✗ | ✗ | ✗ | ✗ | ✗ | ✗ | SVSH |
| **INR** (Philip et al., 2021) | Multi | ✗ | ✓ | ✗ | ✗ | ✗ | ✓ | ✗ | ✗ | Irradiance Maps |
| **IPT** (Azinovic et al., 2019) | Multi | ✓ | ✓ | ✓ | ✗ | ✗ | ✓ | ✓ | ✓ | Emissive Objects |
| **PB-InvIndoor** (Li et al., 2022) | Single | ✓ | ✓ | ✓ | ✗ | ✗ | ✓ | ✓ | ✗ | 4 x SGs |
| **FIPT** (Wu et al., 2023) | Multi | ✓ | ✓ | ✓ | ✗ | ✗ | ✓ | ✓ | ✓ | Emissive Texture |
| **Ours - NL** | Multi | ✓ | ✓ | ✓ | ✓ | ✓ | ✓ | ✓ | ✓ | Learned Local |

Table 1: **Comparison to prior works.** Earlier works focused mostly on virtual object insertion and use incident illumination models, which permits consistent scene relighting. Recent methods aim at physically-based reconstruction together with lighting editing. However, they use hand-crafted emission models and heuristics to constrain their optimization. In our work, we use a learned model with learned geometry-based priors to reconstruct high-quality emissions and constrain the ill-posed problem of appearance decomposition.

optimization. Instead of hand-crafted models, we learn a generic emission model and use learned priors to constrain our optimization, which allows high-quality emission reconstruction even from a sparse set of views. We provide a summary of prior works in Tab. 1.

**Virtual Object Insertion.** Recent image-based rendering methods use single or stereo images to predict incident illumination to shade the object (Gardner et al., 2017; 2019; Wang et al., 2022a; Srinivasan et al., 2020; Wang et al., 2021; Prakash et al., 2019; Li et al., 2020; Zhu et al., 2022; Wang et al., 2022b). They shine in a single-view setting, producing convincing insertion. However, they either use global lighting representation or need to train their network on synthetic data causing domain gap.

Physically-based rendering (Azinovic et al., 2019; Nimier-David et al., 2021; Li et al., 2022), such as ours, can model light transport properly; however, their challenge is to find a good compromise between regularization and expressiveness of the emission model. We utilize learned priors to reconstruct high-quality emissions, which helps the object insertion even near the light sources.

**Relighting.** While a remarkable body of research has concentrated on relighting single objects, room-scale scenes remain a challenging scenario. Indoor Neural Relighting (INR) (Philip et al., 2021) allows for light insertion, but they do not infer the light sources; thus, editing is not possible.

Other methods use inverse rendering to reconstruct mesh light sources (Azinovic et al., 2019; Nimier-David et al., 2021; Li et al., 2022). These methods either require a large set of observations or need to limit the expressiveness of their lighting model to constrain their reconstruction. One key feature of our method is the ability to reconstruct complex emission distributions, even from a sparse set of views and without requiring direct observations of the light sources.

**Neural Fields.** Neural fields have started a new era in 3D scene representation and reconstruction (Xie et al., 2022). Utilizing the expressiveness of neural networks has brought unprecedented quality to appearance reconstruction and novel-view synthesis (Mildenhall et al., 2020). Conditional neural fields have enabled scene manipulation (Park et al., 2019; Sitzmann et al., 2019) and learned priors to constrain reconstruction from a sparse set of views (Sitzmann et al., 2019). Our method further uses explicit inverse rendering to reconstruct the lighting of a scene using learned priors.

## 3 NEURAL LIGHTING PRIORS

In the following section, we present our method. First, we introduce our rendering pipeline (§ 3.1). Second, we describe our scene representation (§ 3.2). Then, we present our rendered dataset (§ 3.3) with training (§ 3.4) and testing (§ 3.5) details. Finally, we describe our voxel-based emitter sampling for noise reduction during rendering (§ 3.6) and show how our method provides control over the reconstructed light sources (§ 3.7). We illustrate our overall pipeline in Fig. 1.

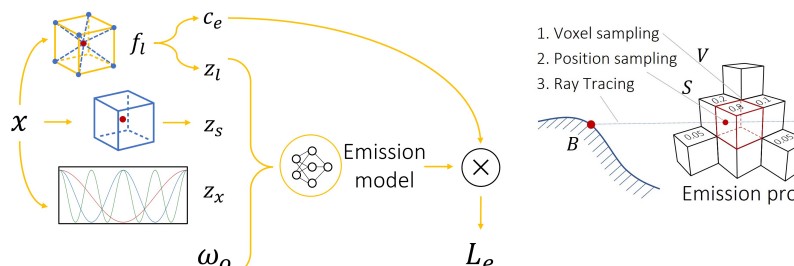

Figure 2: **Emission evaluation.** We show the evaluation pipeline of the surface emission at surface position $x$ in direction $\omega_o$. We apply trilinear interpolation at the lighting voxel grid $G_l$ obtaining lighting features $f_l$. They are split into emission albedo $c_e$ and lighting embedding $z_l$. We also take the nearest semantical embedding $z_s$ from the semantical grid $G_s$. We parameterize our model together with additional positional embeddings $z_x$, and evaluate it with the emission direction as input. Finally, we multiply the predicted emission with the emission albedo to get the final emission value.

Figure 3: **Voxel-based Emitter Sampling.** Emitter sampling is crucial for noise reduction during rendering but requires an explicit lighting representation. We propose Voxel-based Emitter Sampling, where we store an average emission proxy value for each voxel in the scene. First, we sample a voxel $V$ weighted with its proxy value. Second, we sample a point $S$ uniformly inside the voxel. Third, we shoot a ray from the current bounce point $B$ through $S$ and finally, keep the ray if the hit point $H$ is inside $V$.

## 3.1 BACKGROUND

Our goal is to perform inverse graphics, i.e., reconstruct materials and lighting of the scene from image observations. On a high level, we achieve this by inverting the forward imaging process, which is given via the rendering equation (Kajiya, 1986) (Eq. (1)). We consider only surface emissions and surface scatterings without any subsurface interactions. Since this integral is intractable to solve, we approximate it with path tracing using Monte Carlo estimation (Kajiya, 1986). To render a single pixel of the image, we need to estimate the incoming radiance $L_i$ towards the camera. Given a starting position $x_0$, we shoot a ray in direction $\omega$, which hits the scene at position $x_1$. We evaluate the emission $L_e$ at the hit position towards the starting position. To approximate the integral part, path-tracing uses a single sample, i.e., we shoot a single new ray and calculate the scattered radiance given the reflectance $f_r$ and the incident angle $\theta_i$. Then, we estimate the incident radiance recursively. We use Mitsuba 2 (Nimier-David et al., 2019) to implement our path tracer. To generate the camera rays, we use uniform sampling over the pixel area. To sample bounce rays, we use BRDF and emitter multiple importance sampling (Veach & Guibas, 1995), as described in § 3.6. Our work focuses on the emission $L_e$ reconstruction using learned priors given sparse-view observation.

$$
\begin{aligned}
L_i(\boldsymbol{x}_0, \boldsymbol{\omega}) &= L_o(\boldsymbol{x}_1, -\boldsymbol{\omega}) \\
L_o(\boldsymbol{x}_1, \boldsymbol{\omega}_o) &= L_e(\boldsymbol{x}_1, \boldsymbol{\omega}_o) + \\
&\int_\Omega f_r(\boldsymbol{x}_1, \boldsymbol{\omega}_i, \boldsymbol{\omega}_o) \cdot L_i(\boldsymbol{x}_1, \boldsymbol{\omega}_i) \cdot cos\theta_i d\boldsymbol{\omega}_i
\end{aligned}
\tag{1}
$$

## 3.2 REPRESENTATION

**Geometry.** We use explicit triangle meshes obtained in a pre-processing step.

**Lighting.** We propose to represent the surface emission $L_e$ with a locally conditioned neural field $\Theta_l$, as visualized in Fig. 2. The conditioning values are stored in a voxel grid. Our approach combines the expressiveness of a neural field with the explicit representation of a voxel grid, giving us control over the lighting.

However, since our representation is controlled by latent features, explicit control would require conditional training or latent space exploration. Instead, we decompose the surface emission into emission albedo $c_e$ and intensity $I_e$.

$$
L_e = I_e \cdot \boldsymbol{c}_e
\tag{2}
$$

Our neural field $\Theta_l$ predicts the surface emission intensity $I_e$ in direction $\boldsymbol{\omega}_o$ for a given emission distribution defined by a set of local embeddings: semantics $\boldsymbol{z}_s$, lighting $\boldsymbol{z}_l$ and positional $\boldsymbol{z}_x$. The emission direction $\boldsymbol{\omega}_o$ is measured in the local surface-bound frame, and it is positionally encoded. We use the same neural field for all voxels and for all scenes; thus, our model can be seen as a parametric emission distribution.

$$I_e = \Theta_l(\boldsymbol{\omega}_o, [\boldsymbol{z}_s^T, \boldsymbol{z}_l^T, \boldsymbol{z}_x^T]^T) \tag{3}$$

Semantical embeddings $\boldsymbol{z}_s \in \mathbb{R}^{16}$ help to better constrain our model (§ 3.4). Lighting embeddings $\boldsymbol{z}_l$ are stored in a voxel grid $G_l$ of resolution $20cm$. During querying the network, we obtain lighting embeddings $\boldsymbol{z}_l \in \mathbb{R}^{16}$ and emission albedo $c_e \in \mathbb{R}^3$ by trilinear interpolation.

$$[\boldsymbol{c}_e^T, \boldsymbol{z}_l^T]^T = G_l(\boldsymbol{x}) \tag{4}$$

We apply positional encoding on the input position, measured in the local voxel frame to obtain $\boldsymbol{z}_x \in \mathbb{R}^{63}$. We choose the encoding frequencies according to the voxel size to make the encoding continuous over the whole scene.

To restrict the multiplicative ambiguity between the emission albedo $c_e$ and intensity $I_e$, we constrain the albedo to the $[0, 1]$ range during the optimization. However, we found that with a commonly used sigmoid activation, the gradients can easily vanish. Therefore, we propose a new activation function for constrained settings, which we dub Linear Clamp. Our Linear Clamp works as a regular clamp function during the forward. However, during the backward, we keep the gradient if it points toward the valid range. This way, the output range is constrained, and the gradients will also not vanish. Furthermore, we found that our activation also helps to speed up the convergence. For further analysis, we refer to the supplemental.

$$\begin{aligned} y =& \text{LinClamp}(x, x_{min}, x_{max}) \\ =& \min(\max(x, x_{min}), x_{max}) \\ \frac{\partial y}{\partial x} =& \begin{cases} 0 & \text{if } x > x_{max} \text{ and } \partial L/\partial y < 0 \\ 0 & \text{if } x < x_{min} \text{ and } \partial L/\partial y > 0 \\ 1 & \text{otherwise} \end{cases} \end{aligned} \tag{5}$$

**Material.** Inverse graphics requires decoupling the lighting from the material properties. In our work, we focus on the lighting representation and use only a coarse material proxy. We consider only Lambertian materials and represent them with local diffuse albedo values stored in a voxel-grid $G_m$ with a resolution of $10cm$. We use trilinear interpolation to get the diffuse albedo value $\boldsymbol{c}_m$ and constrain it to the $[0, 1]$ range with our Linear Clamp layer.

$$f_r(\boldsymbol{x}, \boldsymbol{\omega}_i, \boldsymbol{\omega}_o) = \boldsymbol{c}_m = G_m(\boldsymbol{x}) \tag{6}$$

## 3.3 DATASET

We aim to learn a generic model of indoor lighting emissions. Learning a prior requires a large dataset. However, to the best of our knowledge, only OpenRooms (Li et al., 2021) provides material and lighting annotations with unbounded HDR renderings of indoor scenes with spatially-varying lighting. Still, we found that their emitters lack complexity for our task. Therefore, we train our model on synthetically rendered observations from the 3D-Front dataset (Fu et al., 2021).

We extend the 3D-Front dataset (Fu et al., 2021) with photo-realistic, physically based lighting descriptions. We found that the defined emitters are not suitable for our task since they are often point light sources and are not defined for all lamps. To get realistic lighting conditions, we define an emission texture for lamp objects, as described in our supplement.

We focus on internal light sources; thus, we do not consider the illumination coming from windows. Therefore, we close the rooms in our dataset by replacing the wall of windows and doors with closed planes. We match the texture of the closed wall to the original. Even though not specifically trained for windows, their emission can be approximated with directional emission profiles; see supplement.

The materials in the dataset are defined as albedo textures with object-specific specularity and roughness values. Our dataset uses the GGX (Walter et al., 2007) microfacet distribution.

For our training set, we prepare 976 rooms, 10 views each. Our test set contains 100 rooms. We render each room from 10 views for optimization and from 10 other views for novel view synthesis evaluation. Since we are focusing on light reconstruction, the first views always look at the light sources. The remaining views are randomly chosen with zero roll, arbitrary yaw, and pitch between 70 and 77 degrees. We also apply the coverage score filtering of BlenderProc (Denninger et al., 2019) to select views with more objects.

## 3.4 TRAINING

**Semantical prior.** Decomposing the appearance into lighting and material parameters is a highly ambiguous problem. To better constrain this task, we introduce a semantical prior for lighting reconstruction. Given the reconstructed geometry of a scene, we predict a voxel grid of semantical features $G_s$ at the same resolution as our lighting embedding grid $G_l$. We use the same semantical embedding $z_s$ for every point in a voxel.

Our semantical prediction model is a binary segmentation network. We use the ScanNet-pretrained Res16UNet34D feature extractor network from (Rozenberszki et al., 2022). We fine-tune the network on our training dataset (§ 3.3) for the downstream task of light source segmentation. We keep the encoder frozen and optimize the decoder and classifier. Our model does not use any color information. Based on the binary prediction, we choose between two learnable codes to get our semantical embeddings.

**Emission prior.** In contrast to any other solutions, our approach does not rely on hand-crafted models but learns a parametric emission model of reasonable lighting conditions directly from observations. We train our neural field in auto-decoder fashion (Park et al., 2019) on views rendered under a large corpus of lighting conditions.

Our network is trained to reconstruct the surface emission of the training scenes. We use ground truth geometry, semantics, and lighting description during training. In each step, we randomly select 8192 pixels from 10 randomly selected views. We shoot one ray uniformly selected from the pixel area. Our network $\Theta_l$ is shared across all voxels and scenes. In each optimization step, we update the network parameters as well as the local lighting features.

Our objective consists of two parts. First, we supervise our network with an emission loss, which is the L2 distance between the predicted $\hat{L}_e$ and ground truth surface emission $L_e$ at the hit points , which is available in our dataset. Second, we use an L2 regularizer on our lighting features $f_l$ with a weight of $w_{lf} = 1e-1$.

$$L_{train} = \|\hat{L}_e - L_e\|_2^2 + w_{lf} \cdot \|\boldsymbol{f}_l\|_2^2 \tag{7}$$

We used the Adam (Kingma & Ba, 2015) optimizer with learning rate $1e-2$, betas $(0.9, 0.99)$, and weight decay $5e-4$. Similarly to the work of Nimier-David et al. (2021), we update only those local latent codes, which were used in the current iteration to avoid unnecessary updates caused by the optimizer's momentum. We train our model for a total of 1000 epochs on a single NVIDIA RTX A6000 GPU, which takes around 4 days.

## 3.5 TESTING

For inference, we follow the auto-decoder framework (Park et al., 2019). We apply test-time optimization on the local lighting and material features, but we keep our trained lighting model frozen. At this stage, we assume to have the reconstructed geometry with a limited number of views given.

We supervise the optimization with an $L2$ reconstruction loss and with an $L1$ regularizer on the predicted emissions, as in Azinovic et al. (2019), with a weight of $w_e = 1e-1$. We select 512 pixels from 10 views and approximate the pixel value with path tracing. For each pixel, we shoot 2048 rays, and we trace one bounce. We found that more bounces are beneficial for the material reconstruction, but for lighting reconstruction, increasing the spp value did not yield better convergence. During optimization, we use only BRDF sampling, but while rendering the final results, we use our emitter sampling technique, described in § 3.6.

Obtaining the gradients with respect to the scene parameters requires backpropagating through the rendering equation. Since this is intractable analytically, we again approximate the gradients with

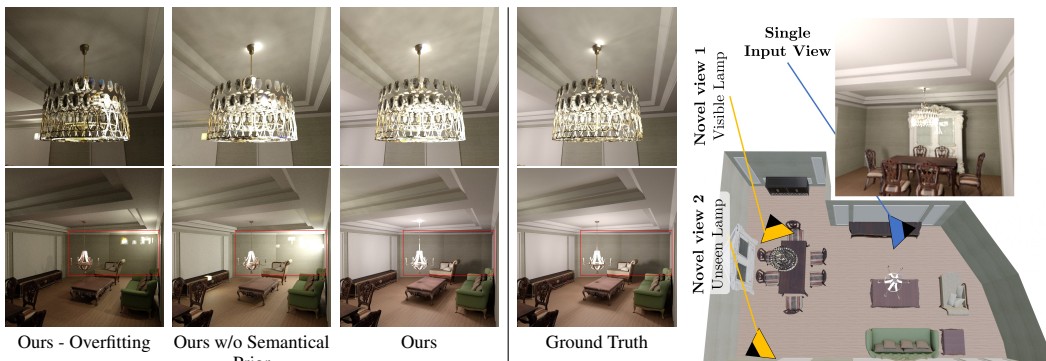

| | Ours - Overfitting | Ours w/o Semantical Prior | Ours | Ground Truth |

Figure 4: **Prior effect ablation.** Given a *single observation* as well as the ground-truth geometry and material properties, we demonstrate that our approach may reconstruct both light sources, one observed in the input image and the other *only observed indirectly*. First, we overfit to the scene and optimize for the emission model parameters together with the local latent codes. Then, we use our emission prior without any semantical information. This prior already constrains the optimization to better reconstruct the visible lamp (top row), but still fails at the unseen lamp (bottom row). Finally, we use the semantical prior, which can properly find the light sources, even if not visible in the view.

| | PSNR ↑ | SSIM ↑ | LPIPS ↓ |
|---|---|---|---|
| Ours Overfitting | 12.61 | 0.534 | 0.335 |
| Ours w/o Semantical Prior | 17.65 | 0.679 | 0.279 |
| Ours | **17.81** | **0.866** | **0.159** |

Table 2: **Prior effect ablation.** Quantitative evaluation of our prior effect ablation (Fig. 4) averaged over 10 test views. Using semantical prior gives important cues about the light sources, but using an emission prior gives further improvement.

Monte-Carlo estimation, similarly to the rendering. Using the same paths as during the rendering would lead to biased estimates, as described in Azinovic et al. (2019). Therefore, we use 2048 new paths for each pixel.

## 3.6 EMITTER SAMPLING

Monte Carlo approximation of the rendering equation yields noisy estimates. To reduce the noise, we apply BRDF and emission multiple importance sampling (Veach & Guibas, 1995). However, emitter sampling requires exact knowledge of the light sources, which we lack. We thus introduce a voxel-based emitter sampling strategy.

We show our sampling strategy in Fig. 3. We store an additional proxy value in our lighting voxel grid $G_l$, which is optimized for the average emission value coming from that particular voxel. In each path tracing iteration, we first sample a voxel $V$ with the weighted probability of the voxel's emission proxy ($p_V$). Second, we uniformly sample a position $S$ inside the voxel. Finally, we trace a ray from our starting point $B$ through the sampled point $S$. If the hit point $H$ is inside the sampled voxel $V$, we keep the ray; otherwise, we discard it. This way, every point along the ray inside the voxel will be mapped to the same surface hit point. Thus the sampling probability is the marginal probability along the ray-voxel intersection ($l$):

$$p = p_V \cdot p_S \cdot l \tag{8}$$

## 3.7 CONTROL

Even though we use a neural emission model, our grid-based local conditioning gives control over the local emission strength and color by changing the emission albedo. Replacing or modifying local lighting features has only local effects. Optionally, one can also compose the reconstructed lighting with additional light sources.

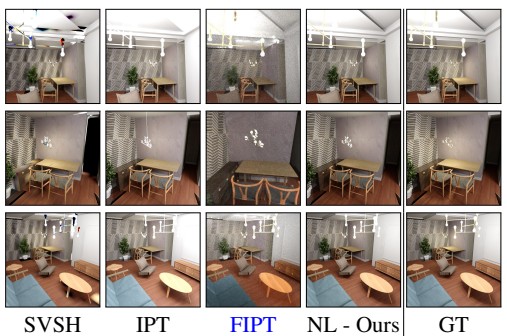

| SVSH | IPT | FIPT | NL - Ours | GT |

Figure 5: **Lighting Reconstruction.** We compare our method against two baselines on the novel-view synthesis task. Given the scene mesh and material, we reconstruct the lighting and evaluate it from novel views. IPT (Azinovic et al., 2019) optimizes a single emission value per object, causing reconstruction artifacts near the light sources. FIPT (Wu et al., 2023) optimizes for more parameters being less constrained, leading to missing emissions in unobserved regions. SVSH (Maier et al., 2017) uses an incident illumination model, which prevents generalization to novel views, while our method reconstructs detailed emissions.

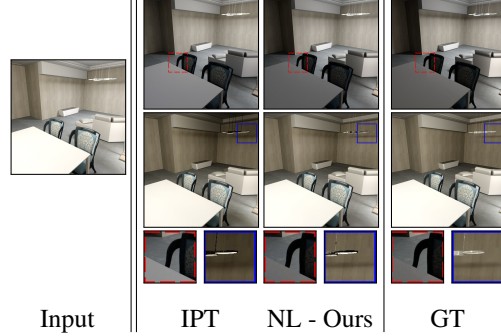

| Input | IPT | NL - Ours | GT |

Figure 6: **Light editing.** We compare our method against IPT (Azinovic et al., 2019) on our light editing benchmark. Given an input scene, we relight the same view by turning off one of the lamps. Since IPT does not reconstruct the light sources perfectly, the relit images contain visible artifacts on the light sources and on the shadows. However, our method reconstructs the light sources precisely, giving favorable relighting.

|  | PSNR ↑ |
|---|---|
| SVSH (Maier et al., 2017) | 22.23 |
| IPT (Azinovic et al., 2019) | 19.25 |
| FIPT (Wu et al., 2023) | 17.59 |
| NL - Ours | **24.89** |

Table 3: Lighting Reconstruction (Fig. 5) averaged over the test views.

|  | PSNR ↑ |
|---|---|
| IPT (Azinovic et al., 2019) | 22.94 |
| NL - Ours | **29.27** |

Table 4: Light Editing (Fig. 6) averaged over the test views.

## 4 EXPERIMENTS

We evaluate our method on synthetic and real indoor scenes. In these experiments, we always fit the specific representation to a set of 10 observations. We use ADAM with method-specific learning rates, as described in the supplementary material. We compare against IPT (Azinovic et al., 2019) and SVSH (Maier et al., 2017), both using low-parametric models, making them better capable of fitting in a sparse view setting. For IPT (Azinovic et al., 2019) experiments, we use a simplified material model similar to ours, i.e., we optimize for diffuse values per object. We visualize and evaluate using 16k spp and tone-mapped renderings, using the transfer function of Kalantari & Ramamoorthi (2017):

$$x \rightarrow \frac{log(1 + \mu x)}{log(1 + \mu)}, \text{where } \mu = 64 \tag{9}$$

**Prior effect ablation.** We ablate our lighting prior in a very sparse setup and show how our learned prior helps to reconstruct even invisible light sources (Fig. 4). Given just a single observation and the ground truth geometry with the materials, we reconstruct the lighting of the scene. In this scene, there are two light sources. One is visible in the input observation, the other is not. First, we do not use any prior but also optimize our emission model, i.e., we overfit to a specific scene. This setting has no notion about reasonable emission distributions and cannot find the second light source. Second, we train our emission model without any semantical information. This way, the trained model just learns an emission prior. This prior helps to better reconstruct the seen light

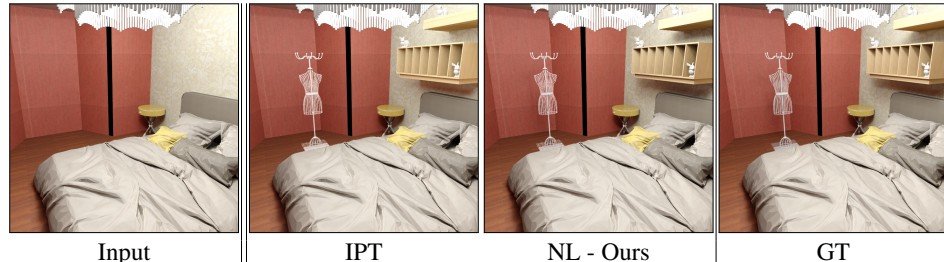

| Input | IPT | NL - Ours | GT |

Figure 7: **Virtual Object Insertion.** We compare our method against IPT (Azinovic et al., 2019) on the virtual object insertion task. We insert additional shelves and a valet into the scene. IPT (Azinovic et al., 2019) cannot reconstruct the light sources perfectly, which causes softer shadows. However, our method produces renderings closer to the ground truth.

|  | PSNR ↑ |
|---|---|
| INR (Philip et al., 2021) | 18,91 |
| NL - Ours | **29.89** |

Table 5: Light Insertion (Fig. 8) average over the test views.

|  | View Synthesis | Virtual Object Insertion Fig. 7 |
|---|---|---|
|  | PSNR ↑ | PSNR ↑ |
| IPT (Azinovic et al., 2019) | 30.23 | 30.08 |
| NL - Ours | **37.72** | **32.00** |

Table 6: Overall scene reconstruction quality on the full test set and virtual object insertion averaged over the test views.

source but does not help in finding the second light source. Finally, we use semantical information, which constrains the optimization well enough to faithfully reconstruct the seen light source and find the second one. We report quantitative results in Tab. 2 averaged over 10 test views, including the reported ones.

**Lighting reconstruction.** We showcase the expressiveness of our approach in Fig. 5. Using ground truth geometry and material, we optimize the local lighting features on a single synthetic scene and evaluate the rerendering in novel views. We compare our representation against Inverse Path Tracing (IPT) (Azinovic et al., 2019) and Spatially-Varying Spherical Harmonics (SVSH) (Maier et al., 2017). Similarly to our method, IPT reconstructs surface emissions but uses a pre-defined cosine emission profile and optimizes only for a single emission value per object, causing errors at the light sources and in the shadows. FIPT is also an optimization-based method; it has no prior about emitters, leading to missing emissions in unobserved regions. SVSH models incident illumination; thus, it has no notion of light transfer. This leads to artifacts on sparsely seen regions, and capturing high-frequency details is limited by the order of basis functions. Our method outperforms the baselines qualitatively and quantitatively (Tab. 3).

**View synthesis.** We benchmark the scene reconstruction quality against IPT (Azinovic et al., 2019). We reconstruct the lighting and material of all the 100 test scenes with both methods and rerender all the 10 fitting views per scene with the same amount of samples per pixel. We outperform IPT with more than $5dB$ (Tab. 6).

**Virtual object insertion (VOI).** Proper lighting reconstruction is crucial for photo-realistic VOI. Even though our representation closely matches surface emissions, there are minor differences. A small difference can have a huge impact on photorealism. To achieve more photo-realistic results, instead of directly using the rerenderings, we use a residual editing, as described in our supplemental.

We benchmark our method against IPT (Azinovic et al., 2019). We insert shelves and one room valet to the scene. IPT (Azinovic et al., 2019) has difficulties with objects closer to the light sources due to improper emission reconstruction. However, our method can faithfully insert the virtual objects and outperform the baseline both quantitatively (Tab. 6) and qualitatively (Fig. 7).

**Light editing.** Our method supports editing the light sources as described in § 3.7. Similarly to the virtual object insertion in § 4, we use residual editing. We compare our method against IPT (Azinovic et al., 2019). We use a scene with two pendant lamps. We fit our method and the baseline to the observations and render the same views under two relit conditions. First, we turn off just the first light source, then vice versa. We relight the scene with both methods and compare the results to ground truth renderings. IPT (Azinovic et al., 2019) archives qualitatively similar results to our method, but it suffers from artifacts close to the light sources due to the wrong lighting

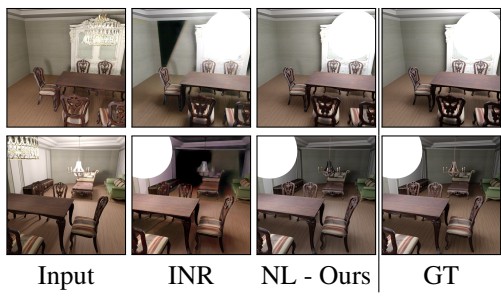

Input          INR        NL - Ours        GT

Figure 8: **Light insertion.** We compare our method against INR (Philip et al., 2021) on our synthetic scene. Given 10 views and the ground truth geometry, we relight the scene by turning off all the light sources and inserting a new spherical light source. INR (Philip et al., 2021) requires a large amount of input views and fails in our sparse setting, and results in missing albedo values, burned-in shadows behind the chair and over-smoothed textures.

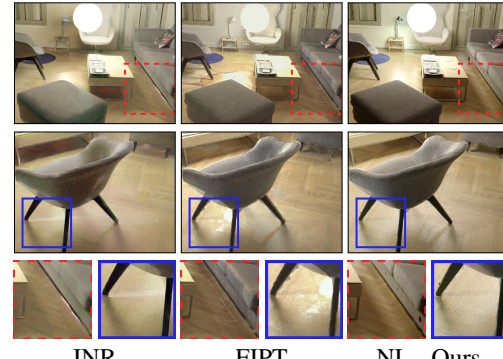

INR          FIPT        NL - Ours

Figure 9: **Real-world light insertion.** We evaluate our method on the real-world scene from Philip et al. (2021) using 10 views. We turn off the lamps and add a new sphere emitter in the middle of the scene. Both INR (Philip et al., 2021) and FIPT (Wu et al., 2023) fail to correctly reconstruct the emissions causing baked-in "white shadows".

reconstruction. Our method produces faithfully relit images close to the ground truth. We show the results in Figure Fig. 6 and in Tab. 6.

**Light insertion.** INR (Philip et al., 2021) is able to turn off all the light sources of the scene and add new ones, but cannot manipulate specific light source. We compare against them in a light insertion setting. Given 10 views and the geometry of a synthetic scene, we relight it by turning off all the light sources and inserting a new virtual spherical emitter. INR (Philip et al., 2021) requires large amount of samples to properly reconstruct the materials; thus, it fails in our sparse setting resulting in missing albedo values, burned-in shadows and over-smoothed textures, while our method can properly relight the scene even from this sparse set of views, as it can be seen in Fig. 8 (Tab. 5).

**Real-world light insertion.** We provide real-world relighting results on the Livingroom scene of Philip et al. (2021) in Fig. 9. We turn off all the light sources and insert a virtual spherical emitter. We compare against INR (Philip et al., 2021) and FIPT (Wu et al., 2023) on a sparse view setup. Directly rendering real inaccurate geometry causes artifacts as for Wu et al. (2023). INR (Philip et al., 2021) alleviates this challenge by using a synthetically trained neural renderer, which gives a smoother surface. Instead, we directly smooth the 3D geometry with Kazhdan & Hoppe (2013) and use physically-based rendering with residual editing (see supplement). A sparse setting brings challenges for both INR (Philip et al., 2021) and FIPT (Wu et al., 2023) and causes incorrect lighting reconstruction with "white shadows". Our method yields favorable results with better shadows and high-frequency details. We show more real-world results in the supplement.

**Limitations.** Our method improves upon the lighting reconstruction quality, but it requires increased computation. Furthermore, we consider emission only from defined surface points. However, our approach can be extended to the more general case, where volumetric emission can also be considered, and the scene can be rendered with volumetric path tracing. Besides, improving the material representation potentially with learned priors is a great avenue for future research.

## 5 CONCLUSION

We introduce Neural Lighting Priors, a learned parametric emission model to better constrain the indoor scene appearance decomposition task given multi-view observations. We have presented an expressive learned lighting representation, which gives control over the reconstructed light sources yet can be fit to unseen scenes with differentiable path tracing. We have also developed a voxel-based emission sampling technique to reduce rendering noise. We have rendered a large-scale synthetic dataset with annotated textured surface emissions and unbounded HDR images to train and test our method. Thanks to our learned priors utilizing semantical information, our model can be fit to a sparse set of views. High-fidelity lighting reconstruction is a key component of virtual and augmented reality applications. We believe that our work takes an important step using learned priors to constrain the ill-posed problem of inverse rendering.

## 6    ETHICS STATEMENT

More realistic virtual representation helps in many real-life problems from robotics to autonomous driving. However, it also makes the virtual world less distinguishable from reality. This can bring problems and make it easier to mislead non-professionals. We believe that to prepare for this effect, we must call society's attention to this danger and show the limits of current technologies as early as possible.

## 7    REPRODUCIBILITY STATEMENT

We describe the rendering algorithm in § 3, including the image formation process, the emitter sampling (§ 3.6) with details in Appendix A.2 and emission evaluation together with the used material representation in § 3.2 and Appendix A.3. The used dataset is detailed in § 3.3 with additional details in the supplementary (Appendix C.1) about the process for generating the emission textures. We define our training procedure in § 3.4 including the hyperparameters and the testing conditions in § 3.5 with additional details about the emitter pruning in Appendix A.4.

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

# Neural Lighting Priors for Indoor Scenes
## — Supplementary material —

In this supplementary material, first, we provide an analysis of our Linear Clamp layer and describe the sampling probability of our proposed voxel-based emitter sampling strategy with further implementational details and discussion in Appendix A. Then, we show additional real-world results in Appendix B. Finally, We describe our experimental setup with our residual editing for photo-realistic compositions in Appendix C.

## A  METHOD DETAILS

### A.1  LINEAR CLAMP

We compare our proposed Linear Clamp activation against the commonly used sigmoid activation. We fit our representation to a scene and report reconstruction PSNR values, as seen in Fig. 10. We apply this layer to constrain the range of the emission albedo $c_e$ of our lighting representation and of the diffuse reflectance value $f_r$ of our material representation. Our proposed Linear Clamp layer does not suffer from vanishing gradients and achieves better final reconstruction with faster convergence.

### A.2  EMITTER SAMPLING

We compare our voxel-based emitter sampling method against pure BRDF sampling in Fig. 11. Using the same number of paths, our approach helps to reduce the rendering noise.

Emitter sampling for Monte Carlo integration requires determining the path probability. We now provide a detailed derivation of the sampling probability described in our paper. Given a bounce point $B$, our goal is to sample a ray $r$ to an emitter and determine the sampling probability ($p(r|B)$). The ray can be determined by its starting position and direction $r = [B, \omega_i]$. Since we need only the closest hit point along the ray, the ray can be reparametrized as the starting and end position $r = [B, H]$. Thus,

$$p(r|B) = p(H|B) \tag{10}$$

Our method samples points in space and not on the surface. Therefore, the ray sampling probability is the marginal probability over all spatial positions along the way, which requires integrating over the whole ray. To simplify the calculations, we apply our fourth step, which rejects every sample outside the sampled voxel. This way, we need to consider only samples inside the voxel of the surface hit point.

$$p(H|B) = \int_{S \in V} p(H|S, B) \cdot p(S|B) dS \tag{11}$$

The position sampling probability $p(S|B)$ can be determined as $p_V \cdot p_S$. Since we used uniform sampling inside the voxel, the position sampling probability $p(S|B)$ does not depend on the position and can be pulled out of the integral. The hit point probability $p(H|S, B)$ is 0 if point $S$ does not lie on the ray and 1 otherwise. Therefore, integrating over the voxel boils down to calculating the voxel-ray intersection ($l$), i.e.,

$$p(H|B) = p_V \cdot p_S \cdot l \tag{12}$$

### A.3  MATERIAL SMOOTHNESS

In our work, we assume the materials to be spatially smooth. We enforce this heuristic prior by applying a total variation regularizer (Rudin & Osher, 1994) on the material grid. We calculate the regularization only for the sampled surface points.

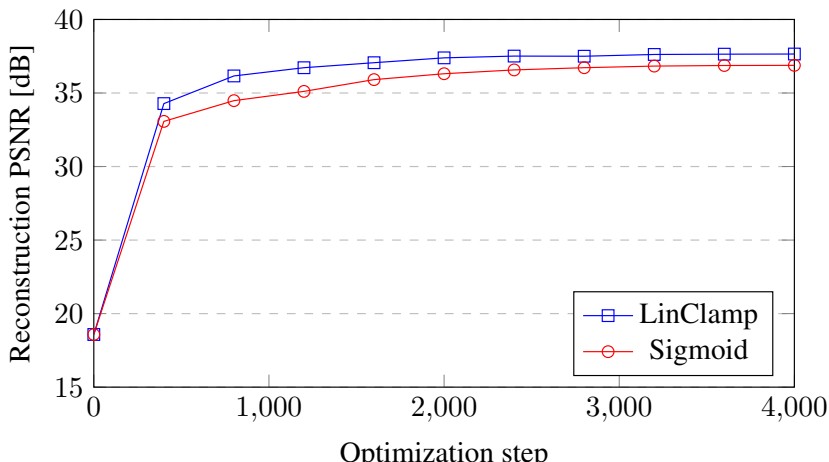

Figure 10: **Linear Clamp.** We analyze the effect of our LinClamp layer. We reconstruct the same scene with our proposed and with sigmoid activation used for constraining the emission albedo $c_e$ and the diffuse material reflectance $f_r$ values. LinClamp achieves better reconstruction and also converges faster.

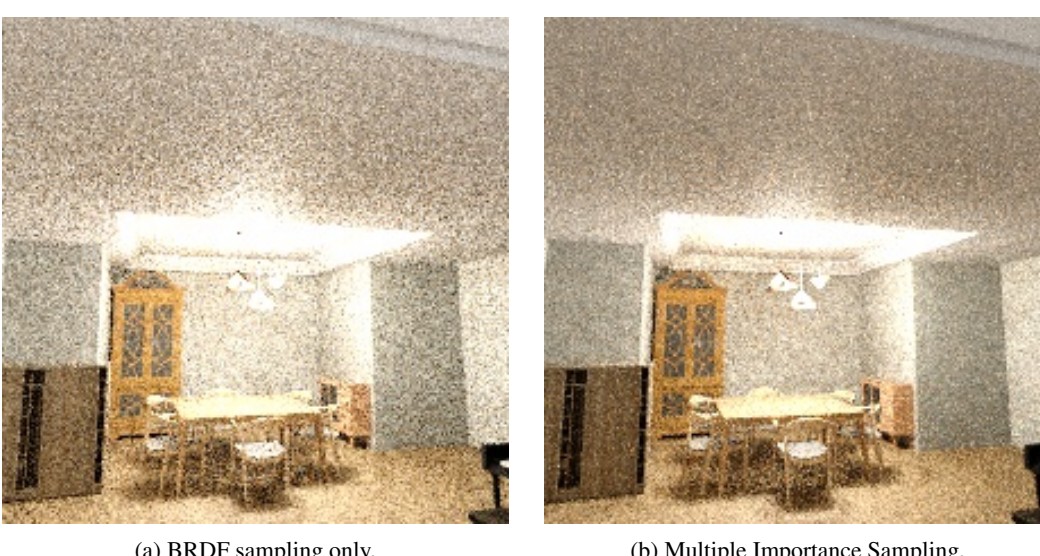

(a) BRDF sampling only.             (b) Multiple Importance Sampling.

Figure 11: **Emitter sampling.** We compare the rendering results with and without our proposed voxel-based emitter sampling using the same number of paths (2048 BRDF paths vs 1024 BRDF + 1024 emitter paths). Our sampling helps to reduce the noise.

## A.4   EMITTER PRUNING

In test time, we have found that the emission regularization helps in the lighting reconstruction, but there can still remain unnecessary emitters. Thus, we apply an emitter pruning technique similar to IPT (Azinovic et al., 2019). After every epoch, we set the emission to zero for every voxel, where the emission proxy value is under $10\%$ of the maximum proxy value.

## A.5   DOMAIN GAP

The biggest challenge in domain transfer is if there is an image encoder trained on synthetic data, which we don't have. The only gap occurs between the real and synthetic emission profiles, which is

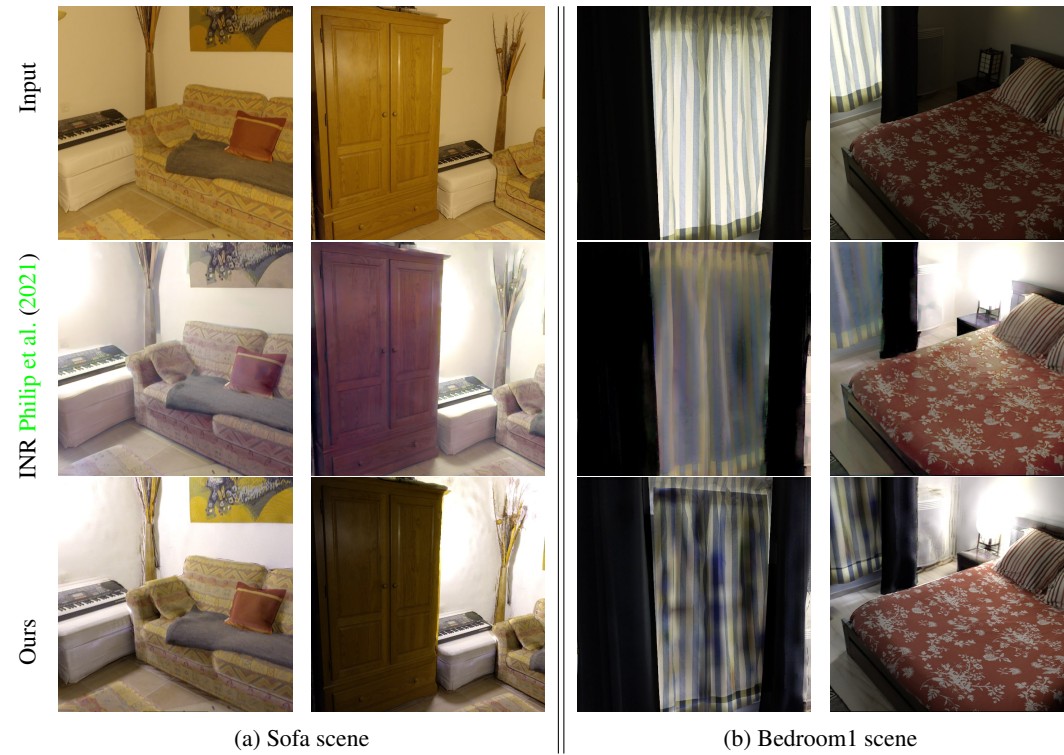

(a) Sofa scene             (b) Bedroom1 scene

Figure 12: **Real-world light insertion** on the scenes of Philip et al. (2021).

much smaller since they are generally smooth and low-dimensional. Thus, domain transfer is easier in our case, as we can see in our real-world examples.

### A.6 SEGMENTATION NETWORK

The goal of our segmentation network is to drive emission optimization but not to rely on it directly. In Fig. 12b, our method successfully assigns emission to the windows even without being segmented as a light source.

## B ADDITIONAL RESULTS

### B.1 REAL-WORLD SCENES

We compare our method against INR (Philip et al., 2021) on two additional scenes of Philip et al. (2021) in Fig. 12. Due to having only a sparse set of 10 views and relying on synthetically trained neural renderer, INR (Philip et al., 2021) gives smoothed results sometimes with incorrect material colors.

### B.2 EMISSION AND BRDF EVALUATION

We show an additional evaluation of the emission and BRDF in Fig. 13.

## C EXPERIMENT DETAILS

During training, we found that loading a room often requires much time and memory, which can become a bottleneck. To overcome this problem, we reuse the same room. During the view sampling, we shuffle the scenes. Then, we load a batch of 10 rooms into the memory. We run 10 epochs over these 10 rooms, then continue with the next batch of rooms.

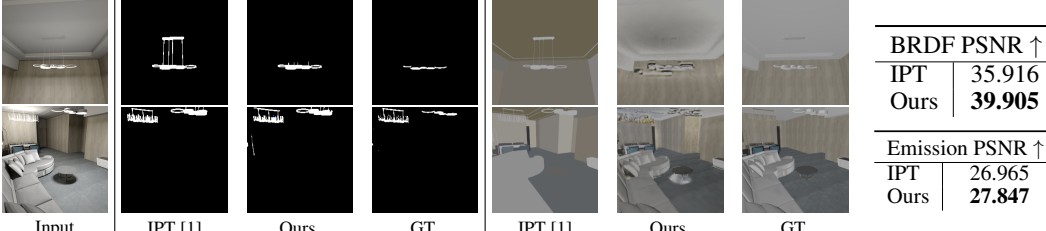

| | BRDF PSNR ↑ |
|---|---|
| IPT | 35.916 |
| Ours | **39.905** |

| | Emission PSNR ↑ |
|---|---|
| IPT | 26.965 |
| Ours | **27.847** |

Input  IPT [1]  Ours  GT  IPT [1]  Ours  GT

Figure 13: **Emission and BRDF evaluations** qualitatively and quantitatively on the test views of the provided scene. Our method outperforms IPT (Azinovic et al., 2019) in both cases. Since our optimization focuses on the lighting and uses 1 bounce rendering, shadowed regions have material artifacts.

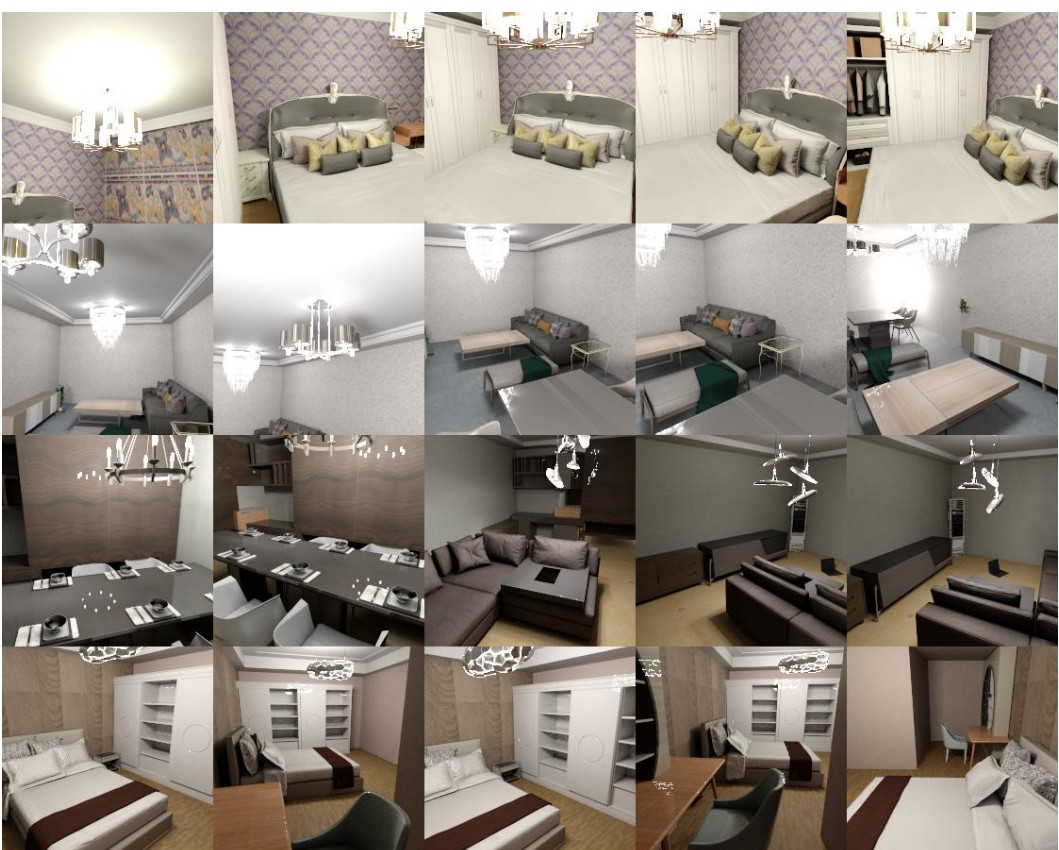

Figure 14: **Dataset.** Example test scenes from our dataset.

During both training and fitting, we apply learning rate schedule. We decrease the learning rate by a factor of 5 after 40%, 70%, and 90% of the total number of epochs. During fitting, we use a single bounce, but when rendering the final results, we use three bounces and 65536 samples per pixel.

## C.1 DATASET

We show example scenes from our dataset in Fig. 14. We use 224x224 resolution for training and fitting, but we render higher resolution (720x720) images for visualization.

We render our dataset with textured mesh emissions from the 3D-Front dataset. To get the emission textures, we apply an adaptive thresholding mechanism. Our adaptive thresholding consists of two main steps. In the first step, we remove the specular highlights baked into the texture. Therefore, we apply our adaptive thresholding technique to find small fragments of bright parts. First, we collect

the brightest 50% of the pixels, measured in L2-norm. We sort the pixel intensities and find the largest gap between two intensity values. Then, we select every pixel above the largest gap. Finally, we apply erosion and dilation operations to remove the small fragments. In the second step, we remove the darker regions and keep only the emissive parts. We apply a similar approach as in the first step, except that instead of considering the brightest 50% of the pixels for the threshold calculation, we drop the darkest and brightest 10%.

## C.2   BASELINE COMPARISONS

**SVSH (Maier et al., 2017).** In the SVSH (Maier et al., 2017) experiments, we use a voxel grid of SH parameters at the same resolution as our lighting grid (20cm). We use second-order approximation, which gives 27 trainable parameters per voxel. We use the same learning rate scheduling strategy as for our method (Appendix C), starting from $5e-1$.

**IPT (Azinovic et al., 2019).** In the IPT (Azinovic et al., 2019) experiments, we optimize for 3-channel emission colors and 3-channel diffuse colors per object. We use the same learning rate scheduling strategy as for our method (Appendix C), starting from $5e-1$. We thank the authors of IPT (Azinovic et al., 2022) for the helpful discussions.

**INR (Philip et al., 2021).** INR (Philip et al., 2021) has been developed to handle lower HDR ranges extracted from raw images, but our synthetic dataset contains unbound HDR samples. To overcome this difference, we increased the INR (Philip et al., 2021) light detection threshold to properly capture the light sources.

For both the synthetic and real-world experiments, we automatically tuned the renderings to best match the ground truth or one selected reference image. The lighting and material properties can be decoupled only up to a global scaling factor due to their multiplicative invariance. Furthermore, INR (Philip et al., 2021) uses a neural rendering approach in their pipeline; thus, it is not ensured that the inserted light sources will keep the emission value after the rendering. Therefore, we tune both the emissions and materials. We determine an overall exposure value required to match the emitter values of the rendered images to the reference and update the whole image. Then, we determine the scaling factor for the materials and scale the non-emissive pixels accordingly.

We thank the authors of INR Philip et al. (2021) for helping in running their method and validating the results.

## C.3   RELIGHTING

We propose to use residual editing to further improve photorealism. We first rerender the view under the original and changed lighting conditions. We calculate the proportional difference between the relit and reconstructed renderings. Finally, we apply this difference to the original input image. We visualize the whole pipeline in Fig. 15.

## C.4   VIRTUAL OBJECT INSERTION

Similar to the relighting, we propose to use residual editing for more photorealistic virtual object insertion. We reconstruct the original view and rerender the same view together with the virtual objects inserted potentially under changed lighting conditions. We calculate the multiplicative difference between the reconstructed and rerendered images. Finally, we apply the difference image to the original view, but we mask the pixel values corresponding to the inserted objects and use the rerendered pixels there.

Similarly, as described in Appendix C.2, a crucial issue with evaluating VOI is that the lighting and material parameters can be optimized only by up to a multiplicative factor. Naively inserting the object into the scene would not ensure that the relative reflectance between the inserted and reconstructed materials matches. Therefore, we tune the reflectance of the inserted objects during our quantitative comparisons. We rerender the view with the virtual objects using a lower number of samples per pixel. We compare the pixel values of the inserted objects to the ground truth and determine an average 3-channel scaling factor. Finally, we multiply the inserted objects' reflectance

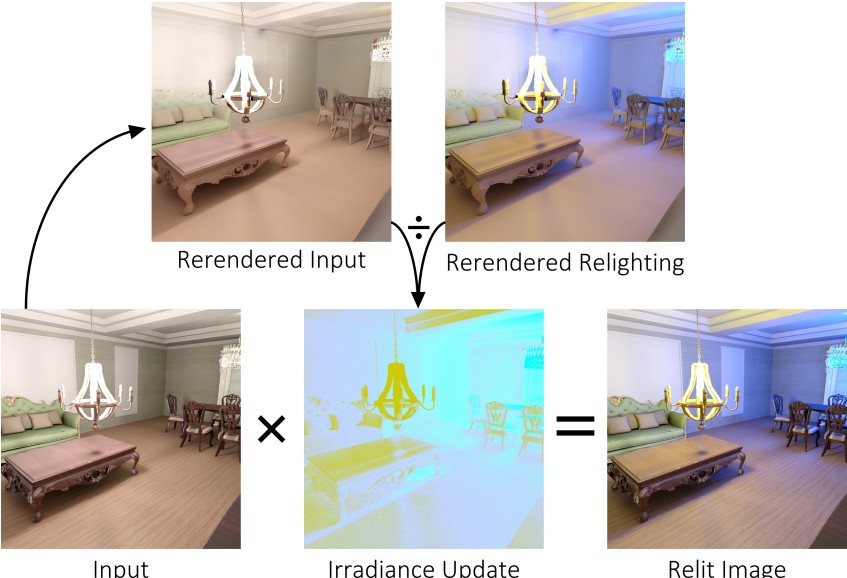

Figure 15: **Residual editing for relighting.** We rerender the scene under the original and manipulated lighting conditions. We estimate an irradiance update on the rerendered images and apply it to the original images.

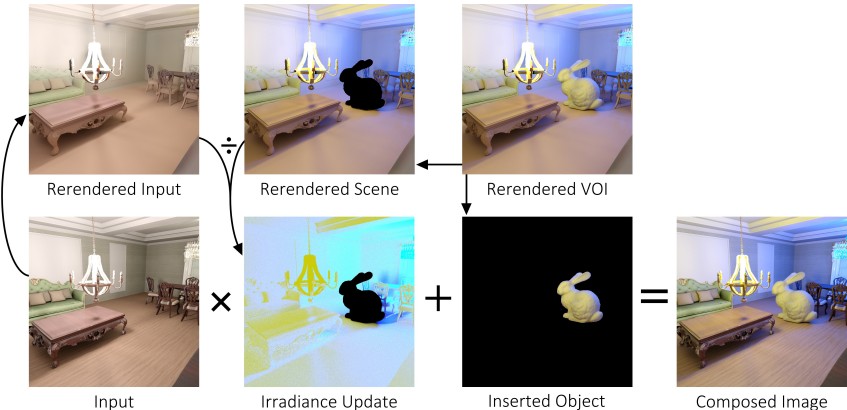

Figure 16: **Residual editing for virtual object insertion.** We follow a similar approach as for our relighting (Fig. 15). However, we directly use the rerendered pixels of the inserted objects and update the irradiance only at the remaining part of the scene.

value with the same scaling factor and rerender the images in higher quality. This way, we can ensure that the inserted objects have the same relative reflectance in the reconstructed scene as in the ground truth scene.

## C.5 RUNTIME

Currently, our method takes ∼70 minutes for real-world fitting, ∼55 minutes for rendering (720x720 resolution, 3 bounces, 65536 spp) on a single A6000 GPU, depending on the scene's complexity. At the same time, our implementation is highly unoptimized and could easily be tuned for speed. We believe that learned priors, denoising techniques, and specialized hardware can improve the runtime.

