# OpenReview forum: "Neural Lighting Priors for Indoor Scenes"
_ICLR.cc/2025/Conference — Submitted to ICLR 2025_

### Official Review · Reviewer_Pyw7 · 2024-10-20

**Soundness:** 2
**Presentation:** 2
**Contribution:** 2
**Rating:** 5
**Confidence:** 5

**Summary:**

The paper proposes Neural Lighting Priors, a method that combines learning and optimization to recover 3D neural surface emission fields from sparse multi-view images and explicit geometry for indoor scenes.

Leveraging explicit geometry, the authors introduce ray tracing to propagate features. They design a learned parametric emission model that decouples spatially varying lighting and material parameters, with a voxel grid storing local latent codes.

During training, both the latent codes and decoder are optimized. In testing, only the latent codes are optimized based on sparse views and geometry. This approach allows the estimation of light sources, supporting accurate shadows and improving photo-realistic relighting and virtual object insertion.

However, the method’s reliance on explicit geometry limits its generalizability.

**Strengths:**

+ Incorporating path tracing:

The use of path tracing allows for the accurate estimation of high-order lighting effects, particularly in generating realistic shadows, which enhances the overall relighting and rendering quality.

+ Learning-based decoder regularization:

The method employs a learned decoder to regularize the latent codes during testing, which effectively reduces overfitting during optimization and leads to more robust lighting reconstructions.

+ Improved performance:

The proposed approach demonstrates superior performance compared to several baselines, particularly in handling sparse-view settings for relighting and virtual object insertion tasks.

**Weaknesses:**

I have several concerns regarding the motivation, implementation details, comparison, and baselines.

+ A little overclaiming the problem setting:

The paper seems to overclaim the difficulty of realistic lighting optimization and relighting from sparse views. While this is undoubtedly a challenging task, the introduction of accurate geometry substantially simplifies the problem. Acquiring geometry is the key aspect of light transport optimization, especially for high-order effects like shadows. However, it is hard to obtain accurate geometry from just 10 sparse views. As a result, this makes the comparison with other methods somewhat unfair, as they don’t have such strong geometry priors. Moreover, how could such a setting be applied to real-world applications? I would encourage the authors to discuss potential use cases, as combining sparse view settings with geometry acquisition seems somewhat unusual.

Additionally, how does the proposed method differ from Nimier-David et al.'s paper on "Material and Lighting Reconstruction for Complex Indoor Scenes with Texture-space Differentiable Rendering," which also uses explicit geometry and path tracing for texture recovery?

+ Writing clarity:

The writing can be significantly improved. It seems the authors focus more on how the task is conducted rather than explaining why the approach was designed this way. Specifically, there are several concerns regarding the implementation:

a) Material model: On lines 247–248, the paper mentions only Lambertian materials, while in line 269, the dataset is described as using GGX. Why are these different, and how do the authors compensate for the material discrepancy?

b) Ray tracing settings: It appears the method uses only 1 bounce and 1 importance sampling (line 195). How is smooth rendering achieved with just one sample? For instance, in Fig. 11, one sample is typically noisy, yet the renderings shown in the supplementary videos look much smoother. In line 319, it is mentioned that during testing, each pixel shoots 2048 rays. This creates confusion between the training and testing settings. Could you clarify this?

c) Emission model: In Eq. (7), is ​$L_e$  the same as the image color? Emission should typically be evaluated only for light sources. Do you use semantic masks to infer these regions?

d) Methodology clarity: Overall, the methodology lacks sufficient detail, and I couldn’t find these details in the supplementary material. I strongly encourage the authors to revise the method section and ensure consistent terminology throughout.

+ Baselines and fairness:

Many of the baselines appear outdated. As I mentioned earlier, the use of geometry priors gives the proposed method a significant advantage over others, which is somewhat unfair as the baselines use geometry inferred from a single image (e.g., Wang et al.). I strongly encourage the authors to include a comparison that does not rely on ground-truth geometry but instead infers geometry from the 10 sparse views, to assess the quality of the inference.

Furthermore, the authors should discuss the relationship between their work and the following papers:

Wang et al., Neural Fields meet Explicit Geometric Representations for Inverse Rendering of Urban Scenes, CVPR 2023: This paper also introduces ray tracing to synthesize shadows.
Wang et al., Neural Light Field Estimation for Street Scenes with Differentiable Virtual Object Insertion, ECCV 2022, This work focuses on inserting objects from a single image.

**Questions:**

See above.

---

> ### Author Response · Authors · 2024-11-22
> **Author Response**
>
> We thank the reviewer for their thorough feedback! We are happy to hear that they appreciated the use of path-tracing and our decoder-only design, which reduces overfitting and leadt to robust lighting reconstruction with superior performance compared to several baselines.
>
> **Problem setting**
>
> The main goal of our paper is to introduce learned priors for the challenging lighting estimation task in a sparse view setting. Although our method requires geometry, which is indeed a challenging, but an orthogonal problem. Sparse-view reconstruction methods as MonoSDF, DiffRF, or Zero123 have already shown great steps in sparse-view geometry reconstruction. We firmly believe that this development will continue and sparse-view geometry reconstruction won't pose a limitation for our method. Therefore, our work focuses on the under-explored sparse-view lighting estimation, which is a challenge on its own, as also shown by our results, how IPT struggles in such a setting.
>
> The work of Nimier-David et al. is an optimization-based inverse rendering method using a dense-view setup, similar to IPT, but focusing on achieving more material details. The key of our method is to use learned lighting priors to enable sparse-view fitting. We will add this discussion to the paper.
>
> **Writing clarity**
>
> We are grateful for the points outlined by the reviewer and will rephrase the respective parts.
>
> (a) We designed our dataset to be as photo-realistic as possible, so that the samples can be closer to real scenes, making our dataset more useful for the community; thus, we used a more complex BRDF model. Since our work focuses on the lighting prior, the difference between the dataset BRDF and our optimized BRDF does not influence the training, when we only consider the emissions. Evaluating with the photo-realistic GGX renderings shows that our method is robust against the material discrepancy.
>
> (b) There are three different settings for our path tracing. First, during training the lighting prior, we use 1 bounce, 1 spp (L.296), since we supervise on emissions directly and have direct access to the ground truth emission values. Second, during fitting we use 1 bounce, 2048 spp (L.319), since we supervise on the rerendering, which should not be noisy. However, often this amount of samples still cannot produce clear results (Fig. 11. in supplemental), but still good enough to provide stable gradients. Third, for evaluations and visualizations in the paper and video, we used a higher spp value (16k) to get clear renderings of our reconstruction. Our description in L.195 is referring to the general rendering process for a single camera ray (1 spp).
>
> (c) During training, we directly supervise on the emission values to avoid rendering noise, which is available in our dataset. This value is zero for non-emitters.
>
> (d) We thank for the suggestion and will update that part.
>
>
> **Baselines and fairness**
>
> To make the comparisons to other methods fair, we always gave access to the geometry (or materials) for the baseline methods as well. We are working on a comprasion with 10-views reconstructed geometry and we are happy to include it.
>
>
> We thank again the reviewer for the insights and we will include a discussion about the mentioned papers!

---

> > ### Comment · Reviewer_Pyw7 · 2024-11-23
> >
> > I thank the reviewers for their detailed explanations. After carefully reviewing the rebuttal and the comments from other reviewers, I have two main concerns and would appreciate further clarification from the authors.
> >
> > 1) Real-world applications. My primary concern relates to the practicality of the proposed approach in real-world scenarios. While I acknowledge that sparse-view reconstruction has shown significant advancements, it appears to still fall short of being a complete solution. For instance, I doubt that MonoSDF can achieve perfect sparse-view scene reconstruction for a complex scene. Meanwhile, methods like DIFRFT or Zero123 focus more on object-level reconstruction. That said, while I understand the authors' argument that geometry acquisition is orthogonal to their contribution, this assumption seems overly strong and may limit the applicability of the paper to real-world use cases. Additionally, I agree with Reviewer KeNN that it would be helpful to explore the performance on dense-view reconstruction as well, as the current comparison may be somewhat unfair.
> >
> > 2) Writing and reproducibility. I encourage the authors to include more detailed explanations throughout the paper. However, the inclusion of three different path-tracing settings adds significant complexity, which undermines the work’s overall reproducibility.
> >
> > Overall, as with KeNN, I would strongly encourage the authors to demonstrate the proposed method in a real world scenario, otherwise, it is very challenging to justify the claim.

---

> > > ### Author Response · Authors · 2024-11-27
> > > **Author Response**
> > >
> > > We thank the reviewer for their quick reply!
> > >
> > > 1. We agree that sparse-view geometry reconstruction is still not solved. Recent methods have shown that using stronger and stronger geometric priors, like monocular priors for MonoSDF, diffusion priors for Zero123 leads to better sparse-view reconstruction. Our goal is to make the first step in the direction of lighting reconstruction. As also shown by our newly provided results, current methods fail in a sparse-view setting due to the lack of a prior.
> > >
> > > We are happy to provide a comparison with dense views in the final revision; however, we would like to note that such a comparison will be highly biased towards FIPT, since dense-view setting is their intended assumption and not ours.
> > >
> > > 2. We have added the discussed details in our latest revision. If the reviewer feels that specific parts are still not well-documented, we are happy to provide more details.

---

### Official Review · Reviewer_skt1 · 2024-11-03

**Soundness:** 2
**Presentation:** 3
**Contribution:** 3
**Rating:** 6
**Confidence:** 3

**Summary:**

The purpose of this paper is to accurately restore lighting in indoor scenes using a limited number of observed views, enabling realistic lighting effects for applications such as virtual object insertion and lighting editing.

To achieve this, a neural-field-based parametric lighting representation is used. This model predicts the intensity and direction of lighting at each location in 3D space and enhances restoration accuracy by limiting possible light source locations through semantic embeddings. Additionally, differentiable path tracing is used to optimize alignment with observed images, and voxel-based sampling is introduced to reduce noise during path tracing and improve efficiency.

The dataset is an extended version of 3D-Front, incorporating Lambertian materials and texture-based lighting to create synthetic data under diverse indoor lighting conditions, which is used for training and evaluation.

**Strengths:**

1. Tackling the Challenge of Lighting Reconstruction in Complex Indoor Scenes

Indoor scenes are particularly challenging for lighting reconstruction due to the variety of light sources, complex geometries, and multiple reflections involved. This paper’s approach appears to address these complexities by enabling accurate lighting reconstruction even with limited observed views, which could potentially broaden the scope for realistic lighting manipulation in intricate indoor environments.

2. Thoughtful Integration of Multiple Techniques

The paper combines neural field-based lighting representation, semantic embedding, differentiable path tracing, and voxel-based sampling in what seems to be an efficient and well-balanced way. This integration could be a strength as it leverages the individual advantages of each technique, allowing for both high-quality reconstruction and computational efficiency in complex lighting conditions. The combination might help enhance the model’s accuracy and robustness, making it better suited to the intricate lighting variations characteristic of indoor scenes.

**Weaknesses:**

1. Reliance on Synthetic Datasets

This paper primarily uses synthetic datasets such as 3D-Front to train and evaluate the model. While synthetic data is useful for demonstrating the feasibility of lighting reconstruction in complex indoor scenes, it may not fully capture the variability and complex material characteristics of real-world environments. Incorporating real-world datasets (e.g., ScanNet, Replica) could help validate the model's generalization performance in actual settings.

2. Assumption of Complete Geometry Conflicts with Limited View Claims

This paper claims to enable lighting reconstruction with limited views, but it is based on the assumption that precise geometry of the entire scene is already available. In an approach intended to work with limited views, assuming pre-existing complete geometry may be logically inconsistent. The notion of limited views generally implies incomplete scene information; however, by assuming full geometry is already provided, the paper may face limitations in fully evaluating the effectiveness of reconstruction with truly limited views.

**Questions:**

1. Obtaining Complete Geometry and Material Information in Realistic Scenarios

In realistic scenarios, obtaining complete geometry and material information for a space requires precise Lidar scanning or highly dense photographic data. Could the authors propose a plausible scenario in which the experiments in this paper could be reproduced in real-world settings with only sparse views to reconstruct the scene’s geometry and material properties?

2. Integration with Modern Radiance Field and Semantic Segmentation Techniques

To achieve a more realistic scene reconstruction, could the authors suggest a scenario in which modern radiance field techniques and compatible semantic segmentation methods (e.g., OpenNeRF) could be integrated with the methodology proposed in this paper?

---

> ### Author Response · Authors · 2024-11-22
> **Author Response**
>
> We thank the reviewer for their valuable insights! We are happy to hear that they found our work a thoughtful, well-balanced integration of multiple techniques, which leverages the individual advantages of neural representations and differentiable path tracting to tackle the challenge of complex indoor lighting reconstruction.
>
> **W1. Reliance on Synthetic Datasets**
>
> We agree that real-world scenes has much higher diversity and the best option would be to train on such data. Unfortunately, large-scale real-world decomposed dataset is not available. To test on real samples, we used scenes from INR as seen in Fig. 9. and also in our supplemental Fig. 12.
>
> **W2-Q1. Assumption of Complete Geometry Conflicts with Limited View Claims**
>
> The main goal of our paper is to introduce learned priors for the challenging lighting estimation task in a sparse view setting. Although our method requires geometry, which is indeed a challenging, but an orthogonal problem. MSparse-view reconstruction methods as MonoSDF, DiffRF, or Zero123 have already shown great steps in sparse-view geometry reconstruction. We firmly believe that this development will continue and sparse-view geometry reconstruction won't pose a limitation for our method. Therefore, our work focuses on the under-explored sparse-view lighting estimation, which is a challenge on its own.
>
> **Q2. Integration with Modern Radiance Field and Semantic Segmentation Techniques**
>
> Our method is based on mesh representation and uses path tracing for rendering. The idea of learning lighting prior could be adapted to radiance fields; however, it would drastically increase the runtime to use volumetric path tracing. One possible option can be to use a sparse radiance field representation, such as 3D Gaussian Splatting, to speed up.
>
>
> We thank again the reviewer for their insights!

---

> > ### Comment · Reviewer_skt1 · 2024-12-03
> >
> > Dear Author,
> >
> > After reviewing other comments, I see that my opinion regarding the "Assumption of Complete Geometry Conflicts with Limited View Claims" aligns to some extent with those of other reviewers. However, I also agree with your point that illumination estimation and geometry estimation are orthogonal problems.
> >
> > In summary, while the combination of various methodologies presented by the authors has demonstrated positive effects, it seems evident that the experimental setting presented by the authors is a very specific environment assumed. For real-world application, it appears highly dependent on sparse-view geometry reconstruction.
> >
> > Nonetheless, effectively demonstrating the validity of combining semantic segmentation, Neural emission model estimation, and a voxel-based method is a valuable contribution.
> >
> > Therefore, I maintain my evaluation.

---

### Official Review · Reviewer_AWrW · 2024-11-04

**Soundness:** 3
**Presentation:** 3
**Contribution:** 3
**Rating:** 6
**Confidence:** 4

**Summary:**

This paper introduces a learned surface emission model to reconstruct 3D lighting fields from sparse multi-view images for indoor scenes.

Leveraging neural fields and voxel-based representations, it enhances relighting and virtual object insertion capabilities. The approach combines explicit inverse rendering with neural priors, enabling complex emission profiles from limited input data by conditioning on 3D spatial and semantic features. This model’s ability to control light sources while requiring fewer views than previous methods suggests advancements in realistic indoor scene applications, such as augmented reality.

**Strengths:**

This paper presents an original and practical approach by combining neural fields with learned priors for indoor lighting reconstruction, allowing it to infer unobserved light sources. The method demonstrates high-quality lighting reconstruction with substantial quantitative gains over previous methods, proving useful for dynamic lighting applications like AR and VR. The inclusion of a voxel-based emitter sampling technique also improves noise reduction, enhancing the overall quality of relighting and virtual object insertion.

**Weaknesses:**

The approach is computationally intensive, requiring significant processing time due to path tracing, which may limit its feasibility for real-time applications. Additionally, the method assumes Lambertian materials, which restricts its adaptability to scenes needing complex material representations, and some of the technical explanations, especially around voxel representation, could benefit from greater clarity.

**Questions:**

- Could you provide more insights into computational optimizations that could reduce the processing time for practical deployment?
- How well does the model generalize to scenes with mixed lighting types? (naturel or artificial)
- Could the approach be adapted for outdoor scenes, where volumetric light transport might play a significant role?

**Details Of Ethics Concerns:**

X

---

> ### Author Response · Authors · 2024-11-22
> **Author Response**
>
> We thank the reviewer for their valuable feedback! We are happy to hear that they found our work original, demonstrating high-quality lighting reconstruction proving useful for dynamic lighting applications and that they appreciate our voxel-based emitter sampling.
>
> **Computational optimization**
>
> One first step in computational optimization could be to optimize our current implementation, which is currently highly unoptimal. Furthermore, the techniques of real-time path-tracing could also be applied in our case, and a coarse-to-fine optimization scheme could also help.
>
> **Generalization to natural lighting**
>
> Since our learned prior does not include the color of the emitter, we can generalize well to the full-range of possible emission colors, including natural light sources as well. One example for this can be seen in our supplementary material Fig.12.b.
>
> **Outdoor scenes**
>
> We believe that the idea of learning a prior for the lighting to better constrain the task could be adapted, but the prior should be probably learned on other data.
>
> We thank again the reviewer for their insights and we will rephrase the description of the voxel representation for greated clarity.

---

### Official Review · Reviewer_KeNN · 2024-11-05

**Soundness:** 2
**Presentation:** 2
**Contribution:** 2
**Rating:** 3
**Confidence:** 5

**Summary:**

The paper proposes a learned surface emission model for indoor scenes, given sparse observations of the scene and ground truth geometry. The model is conditioned on learned semantic features as well as lighting features learned from a large collection of scenes. A test-time optimization of emission and spatially-varying materials is applied, to finetune the features on sparse scene observations, with view synthesis loss and regularization. The paper is trained on synthetic scenes adapted from 3D-FRONT and tested on both synthetic and real-world scenes.

**Strengths:**

[1] Novelty in model design. The paper takes inspirations from several previous works to design a model model which leverages learning-based priors for initial prediction, yet allows for test-time optimization (NeRF-like architecture conditional on grid features) for refinement with physics-based rendering formula. Thanks for those considerations, the method enables a wide range of applications including lighting editing & scene relighting, virtual object insertion & relighting, and produces decent results with sparse view inputs.

[2] The paper is generally well written, with details extremely well-documented. The evaluations are extensive, including both qualitative and quantitative results on numerous inverse rendering tasks, and demonstrate noticeable improvement over multiple prior arts.

[3] The paper also proposes a dataset as an extension to 3D-FRONT, with photorealistic physics-based lighting, which will be valuable to the community once released.

**Weaknesses:**

**[1] Experimental setting with GT geometry.**

The biggest concern regards the setting of the paper. The paper assumes ground truth geometry, the acquisition of which in indoor setting is a challenging problem by itself, especially in sparse-view setting where the paper claims benefits from its learned priors. Without demonstrating results in cases of **estimated** geometry (especially with sparse views, e.g. with methods like MonoSDF), it is difficult to predict how the method will behave with estimated geometry. Previous works (e.g. FIPT, IRIS) have shown it is entirely possible to estimate mesh-based geometry based on dense view inputs and optimize in a similar paradigm to this paper, and works like MonoSDF have shown the possibility of reconstructing fine geometry with sparse-views. As a result, it is entirely possible and essential for the paper to clarify and evaluate in this setting.

**[2] Outdated and incomplete comparison.**

Discussion and comparison on some of the most recent works are missing or incomplete, notably FIPT which is a strong baseline (discussion and comprehensive comparison is doable); MILO which is a baseline from FIPT. Code base and datasets (including both synthetic and real-world scenes) are publicly available from early 2023 by FIPT. Why is comparison against FIPT only provided in Fig. 9 and not in other comparison, given the two methods are directly comparable and FIPT is the SOTA baseline (over IPT).

**[3] BRDF representation.**

The BRDF model is limited to Lambertian diffuse model in this paper, as opposed to account for high-frequency specular lighting as in FIPT, MILO and IRIS. Although the diffuse assumption might work as a good regularization in the presence of strong specular effects VS sparse views (which is a likely explanation for artifacts from FIPT and INR in Fig. 9), without modeling specular component it is not possible to recreate **any** specular effects in downstream applications, and will likely break the view-synthesis objective in scenes with complex lighting. It is crucial for the paper to clarify on the choice, and experiment with additional of specular component to its BRDF representation for a refreshed comparison in Fig. 9.

**[4] Insufficient ablation.**

(a) With/Without test-time optimization. The method depends on test-time optimization to finetune emission and estimate BRDF for to a specific scene, starting from pre-trained features. It will be interesting to observe how the proposed method behaves with and without test-time optimization, especially considering the sparse/single view setting might cause issues like overfitting in optimization.

(b) Path-tracing in sparse view setting. It is not described how path tracing is implemented with sparse-view/a single view. Do we assume complete geometry with sparse-view observations? If yes how is this a practical assumption. Moreover, with limited view coverage and limited number of paths, it is not clear about how the BRDF and emission look like especially in little/not observed regions.

(c) BRDF estimation quality. As described above, considering the method recovers both BRDF and emission, no BRDF results and comparisons are included. It is of crucial importance to observe the results from a physically based inverse rendering method and conclude how the method behaves in the setting of sparse views, considering IPT heavily relies on dense views to reduce biases and promote spatial consistency.

(d) Emission estimation quality. Similar to mentioned above, no separate results on emission estimation/emission masks are provided. Those results are crucial to understand whether the model is able to correct estimate large emission values for emitters and suppress emission for non-emitters, as is done in the IPT paper.

(e) More results on complex scenes and real-world scenes. In case of scenes with complex lighting, geometry and materials (e.g. cluttered furniture, materials with complex patterns and appearance heavily coupled with shadow/highlights, as well as numerous small lamps, windows), and more importantly real-world scenes, evaluation needs to be done to demonstrate the effectiveness of the proposed in challenging and diverse indoor environments. Real-world scenes are readily available for FIPT and has become a popular benchmark in recent works along this line (e.g. IRIS).

**[5] Extra clarification needed.**

(a) in Fig. 9: why do three methods use different geometry? What happens if all use the same geometry, and same BRDF representation (by removing specular term from FIPT, or adding specular to this method). Otherwise it is not a fair comparison.

(b) Modifying local lighting features for light editing. Given the method output no notion of emitters, but dense emission, the claim to 'control over the emitters' might be questionable as the method relies on heavily post-processing to mask out the emitters, and re-render image with edited lighting. What happens if for example emission estimation does not perfectly agree with the emitters, or include false positive emitters (non-emitter regions of high emissions). Do types of those failure cases occur, and if yes how they are handled? Overall, section 3.7 is confusing: 'Replacing or modifying local lighting features has only local effects', which presumably is not desired in light editing?

**[6] Language issues.**

Scene reconstruction -> View synthesis. Table 1: what is geometry prior (it is referring to the learned 3D features?)

**Additional References**

- IRIS: Lin, Zhi-Hao, et al. "IRIS: Inverse Rendering of Indoor Scenes from Low Dynamic Range Images." arXiv preprint arXiv:2401.12977 (2024).
- MILO: Yu, Bohan, et al. "Milo: Multi-bounce inverse rendering for indoor scene with light-emitting objects." IEEE Transactions on Pattern Analysis and Machine Intelligence 45.8 (2023): 10129-10142.

**Questions:**

Please see the above comments for questions to address.

---

> ### Author Response · Authors · 2024-11-22
> **Author Response**
>
> We thank the reviewer for their thorough evaluation! We are happy to see that they found our model design novel (S1), the paper well-written with extensive evaluation (S2) and our proposed dataset valuable (S3).
>
> **[1] Experimental setting with GT Geometry.**
>
> The main goal of our paper is to introduce learned priors for the challenging lighting estimation task in a sparse view setting. Although our method requires geometry, which is indeed a challenging, but an orthogonal problem. MonoSDF and also sparse-view reconstruction methods as DiffRF, or Zero123 have already shown great steps in sparse-view geometry reconstruction. We firmly believe that this development will continue and sparse-view geometry reconstruction won't pose a limitation for our method. Therefore, our work focuses on the under-explored sparse-view lighting estimation, which is a challenge on its own, as also shown by our results, how IPT struggles in such a setting.
>
> **[2] Outdated and incomplete comparison.**
>
> We thank the reviewer pointing out these missing evaluations. We started with evaluating FIPT on our light reconstruction task, as done in L.456 (Tab.3, Fig. 5). FIPT achieves 17.59dB, compared to 19.25dB of IPT and 24.89dB of ours. You can find visual results [here](https://ibb.co/6RW7L9m). Since FIPT is a pure optimization-based method, it has no prior about emitters, leading to missing emissions in unobserved regions. IPT has an object-based prior, i.e. assumes the same emission over an object, which highly reduces the number of parameters to optimize; thus, giving better generalization in sparse-view lighting estimation. We are happy to include a full comparison in the final revision.
>
> **[3] BRDF representation.**
>
> We agree that using a diffuse BRDF is a limitation of our work and that this choice acts as a regularizer. However, we would like to highlight that the baseline methods (INR, FIPT) are failing also on completely diffuse scenes (Fig. 8., upcoming results from above [2]), not being able to correctly find unobserved light sources while our lighting prior helps (Fig. 4).
>
> **[4] Insufficient ablation.**
>
> (a) We thank the reviewer for the suggestion and we are happy to include an ablation on the test-time optimization.
>
> (b) Yes, we are using complete geometry with sparse observations, as also discussed above [1]. Fig.4. shows an example, how the emission is reconstructed in unobserved regions, we are happy to include results about the BRDF of unseen parts.
>
> (c-d) The main focus of our project is the lighting estimation and our method is agnostic to the choice of BRDF; therefore, we haven't evaluated the BRDF directly. We will include a thorough evaluation, but for now we have evaluated the test views of the scene from Fig. 6. Our method outperforms IPT in both BRDF (39.91dB vs 35.92dB) and Emission quality (27.85dB vs 26.97dB), as can be seen [here](https://ibb.co/kqTYCns).
>
> (e) We thank the reviewer for the suggestions and will include scenes from FIPT.
>
> **[5] Extra clarification needed.**
>
> (a) In Fig. 9. all the methods are using the same geometry, which is the geometry used in INR and applied smoothing as post-processing. Our new experiments above [2] compare against FIPT using the same BRDF. We are happy to include FIPT results without specular terms.
>
> (b) In section 3.7 under "local effects" we mean local effects on the emissions, but in rendering it gives global effects. We thank the reviewer pointing this out and we will rephrase it.
>
>
> We thank again the reviewer for the insights and we will incorporate the suggested changes!

---

> ### Comment · Reviewer_KeNN · 2024-11-23
>
> I would like to thank the authors for providing the feedback.
>
> [1] On the results, it would be beneficial to have additional results as promised by the authors. However, due to the significant amount of results promised and not fully provided in the rebuttal, it is difficult to determine the final landscape of comparison and conclude how the results justify the claims.
>
> The link to the FIPT results is broken. It is possible to update the manuscript and supplementary results to includes the results instead of using third-party services. Complete results and comparison need to be added to figures and tables of the paper in an updated version.
>
> [2] On the assumption of complete GT geometry with largely incomplete views, this is an impractical assumption in real world applications. Despite the claim to focus on the inverse rendering instead of geometry estimation, works like FIPT have shown geometry reconstruction for real-world scenes is practical and can be evaluated. Early works like IPT rely on GT geometry to focus on the differentiable path tracing framework, but given the advances of indoor geometry reconstruction between 2019 and 2024, it is fair to conclude that reconstructed geometry with dense/semi-dense/sparse views is a more practical and achievable setting to work on.
>
> The paper mentioned that FIPT failed in cases of unobserved emitters. It is breaking the dense-view assumption of FIPT so that the failure is expected. **A more important question to ask is, what does the comparison look like with dense views, where all of the method will work within their intended assumptions?**
>
> Additionally, assuming diffuse materials for indoor scenes is outdated and impractical. It should be straightforward to extend to complex BRDFs and observe the updated results.
>
> [3] On the learned emission prior, in case of unobserved emitters, does the learned model only rely on textureless geometry features to regress to emission features?

---

> > ### Author Response · Authors · 2024-11-27
> > **Author Response**
> >
> > We thank the reviewer for their quick reply!
> >
> > [1] We are working on the promised results and added the currently finished ones to the paper (Fig.5., Tab.3., Supp.B.2, Supp.Fig.13.).
> >
> > [2] We agree with the reviewer that inverse rendering can be coupled with geometry reconstruction, as FIPT has also shown. However, FIPT assumes dense photometric capture. As also shown in our newly provided results, FIPT fails in a sparse-view setting, even if full geometry and material is given.
> >
> > We are happy to provide a comparison with dense views in the final revision; however, we would like to note that such a comparison will be highly biased towards FIPT, since dense-view setting is their intended assumption and not ours.
> >
> > Although we haven't tried with other BRDFs, simply extending it to a more complex one would make the problem of sparse-view inverse rendering more underconstrained by introducing more parameters to optimize for. Therefore, we believe that similarly to our proposed lighting priors, material priors should be introduced to successfully make that step (L.528.), which we found out of the current scope of lighting priors.
> >
> > [3] Our emission prior predicts intial emission features based on the geometry (L.287).

---

### Meta-Review · Area_Chair_zkZW · 2024-12-23

**Metareview:**

The paper presents a learned emission model for indoor scenes, which estimates diffuse material and spatially-varying lighting, given known geometry. The paper is well-presented and demonstrated good results with sparse input views. One of the main limitations of the paper is the assumption of known geometry, which is important since incomplete reconstructions pose a key challenge to light source estimation and editing. The assumption of diffuse material is also a limitation that prevents specular editing applications. These are significant, since prior works do operate under the more general assumptions of reconstructed geometry and microfacet BRDF models. Lack of comparisons to recent works like FIPT are brought up by multiple reviewers and while the author rebuttal on effectiveness in sparse settings is understood, demonstration of comparisons in various types of conditions including single-view such as Li et al. 2022, sparse such as the proposed work and dense coverage such as FIPT will lead to improved understanding. Overall, the paper can improve in terms of more realistic assumptions and better experimentation, where the authors are encouraged to incorporate the reviewer suggestions towards submission to a future venue. The paper is not recommended for acceptance at ICLR.

**Additional Comments On Reviewer Discussion:**

KeNN notes that assumptions made in the paper over-simplify the problem and prevent real-world application, while also pointing out lack of comparisons to recent works. While the authors respond to these, the reviewer remains unconvinced given the lack of all the required results. AWrW leans to accept and their question on computational detail is addressed by the rebuttal. While skt1 also leans to accept based on the proposed contributions, similar concerns as KeNN are expressed. Pyw7 leans towards rejection, while also noting that the assumption of ground truth geometry limits real-world applicability and comparisons required by reviewers should be included. Overall, the reviewers suggest several directions that the authors may consider towards future improvements.

---

### Decision · Program_Chairs · 2025-01-22

Reject